# Predicting Treatment Response in Inflammatory Bowel Diseases: Cross-Sectional Imaging Markers

**DOI:** 10.3390/jcm12185933

**Published:** 2023-09-12

**Authors:** Irene Mignini, Rossella Maresca, Maria Elena Ainora, Luigi Larosa, Franco Scaldaferri, Antonio Gasbarrini, Maria Assunta Zocco

**Affiliations:** 1CEMAD Digestive Diseases Center, Fondazione Policlinico Universitario “A. Gemelli” IRCCS, Università Cattolica del Sacro Cuore, Largo A. Gemelli 8, 00168 Rome, Italy; irene.mignini@gmail.com (I.M.); rossella.maresca12@gmail.com (R.M.); franco.scaldaferri@policlinicogemelli.it (F.S.); antonio.gasbarrini@unicatt.it (A.G.); mariaassunta.zocco@policlinicogemelli.it (M.A.Z.); 2Dipartimento di Diagnostica per Immagini, Radioterapia Oncologica ed Ematologia, Fondazione Policlinico Universitario “A. Gemelli” IRCCS, Università Cattolica del Sacro Cuore, Largo A. Gemelli 8, 00168 Rome, Italy; luigi.larosa@policlinicogemelli.it

**Keywords:** inflammatory bowel disease, precision medicine, treatment response, cross-sectional imaging

## Abstract

Therapeutic options for inflammatory bowel diseases (IBD) have largely expanded in the last decades, both in Crohn’s disease and ulcerative colitis, including multiple biological drugs targeting different inflammation pathways. However, choosing the best treatment and timing for each patient is still an undeniable challenge for IBD physicians due to the marked heterogeneity among patients and disease behavior. Therefore, early prediction of the response to biological drugs becomes of utmost importance, allowing prompt optimization of therapeutic strategies and thus paving the way towards precision medicine. In such a context, researchers have recently focused on cross-sectional imaging techniques (intestinal ultrasound, computed tomography, and magnetic resonance enterography) in order to identify predictive markers of response or non-response to biologic therapies. In this review, we aim to summarize data about imaging factors that may early predict disease behavior during biological treatment, potentially helping to define more precise and patient-tailored strategies.

## 1. Introduction

Inflammatory bowel diseases (IBD), synthetically classified into the two major entities of Crohn’s disease (CD) and Ulcerative Colitis (UC), are actually complex chronic disorders characterized by a wide heterogeneity in their clinical manifestations and disease course, ranging from mild flares to potentially life-threatening complications. The introduction of infliximab in 1998, the first monoclonal antibody to be approved for CD, revolutionized the treatment approach to IBD [1] and, since then, the therapeutic armamentarium has largely expanded. Currently, in addition to conventional therapies with corticosteroids, aminosalycilates, and immunomodulators, patients with moderate or severe disease may benefit from a growing number of biologic drugs or small molecules [2,3].

However, establishing the best strategy for each patient still represents one of the hardest challenges for IBD physicians. Although derived from advanced technologies, biologics do not grant the same efficacy in all subjects, and some patients experience primary failure or secondary loss of response, needing dose optimization or different therapeutic options. In the absence of reliable predictive markers, physicians are actually obliged to choose among available biologics in an empiric manner, and strict monitoring of the disease course is crucial to confirm or modify therapeutic algorithms. Current management of patients receiving biologics—known as the “treat-to-target strategy”—is based on tight control and frequent reassessments of disease activity in order to promptly detect recrudescence and adapt therapeutic programs to achieve deep remission [4]. Although currently the most accurate approach, such a strategy may require time before identifying the most appropriate treatment for each patient, potentially entailing a prolonged disease burden and high costs for the health system.

Predicting the response to biologic drugs, therefore, would be of utmost importance to directly choose the most appropriate therapy for the single patient, helping to further improve clinical care and cost-effectiveness. Thus, in recent years, physicians’ attention has been primarily directed toward individualized treatments, pursuing an approach based on so-called precision medicine [5]. The European Crohn’s and Colitis Organization (ECCO) has highlighted the key role of precision medicine in IBD in a recent scientific workshop aiming to provide an updated overview on this specific topic. ECCO underlines how clinical features have a limited capacity for guiding clinicians’ choices and that there is an unmet need for more objective predictive biomarkers [6]. It is therefore easy to understand why, in the last few years, studies investigating potential predictors of disease course or treatment response have flourished. Many predictive markers have been examined among clinical, serologic, genetic, endoscopic, and radiologic factors [5]. A graphic synthesis of the treat-to-target and precision medicine approaches is represented in Figure 1.

Cross-sectional imaging, including intestinal ultrasound (IUS), magnetic resonance imaging (MRI), and more specifically magnetic resonance enterography (MRE) and computed tomography enterography (CTE), are essential tools for disease activity monitoring and early detection of signs of response or failure to biological therapies. These techniques provide, in fact, complementary information to endoscopy, allowing one to examine the entire length of the gastrointestinal tract and visualize intramural and extramural lesions. Such capability appears to be of great interest, particularly after the introduction of transmural healing (TH) as an additional therapeutic goal in IBDs. According to the Selecting Therapeutic Targets in Inflammatory Bowel Disease II initiative (STRIDE-II), endoscopic healing is still the main long-term target, leading to treatment changes if not achieved. However, TH, together with histologic remission, has been included as an adjunctive goal to reach deeper healing, especially in CD [4]. Indeed, the role of cross-sectional imaging in the treat-to-target strategy is widely recognized. Nancey et al. recently published a review on the usefulness of cross-sectional imaging (mainly IUS and MRE) to guide physicians’ decisions. They describe how these techniques are accurate in assessing disease activity, extent, severity, and complications. They also underline the crucial importance of identifying imaging markers able to predict disease course [7]. Based on such considerations, research has begun to investigate the utility of cross-sectional imaging not only in monitoring but also in predicting pharmacological treatment response. New imaging markers and score systems have been proposed and examined.

In this review, we summarize the most up-to-date evidence about cross-imaging features having a role in predicting treatment response or failure. We have created three separate sections for the three main imaging techniques, i.e., IUS, MRE, and CTE. In particular, we first focus on IUS, which is widely spread, scarcely invasive, and has a larger amount of available data. We analyze in two different subsections the results of CD and UC. Subsequent sections address MRE, a promising tool in this context, and CTE, which is mainly used in emergency settings and therefore less investigated for disease monitoring and response prediction. Evidence on recent techniques, such as US contrast agents and radiomics, is highlighted throughout the text.

## 2. Intestinal Ultrasound

IUS is a useful, economical, and non-invasive method for assessing and monitoring IBD [8]. Nowadays, it is recommended in combined guidelines by the ECCO and the European Society of Gastrointestinal and Abdominal Radiology (ESGAR) as a sensitive and specific tool for diagnosis, monitoring, and detecting complications in IBD, as well as MRE and CTE [9].

Although IUS is less accurate in exploring some parts of the intestinal tract, especially the proximal small bowel, it has significant advantages compared to MRE in terms of greater availability, lower costs, minor invasiveness, and greater patient acceptability. Indeed, IUS does not require fasting or intravenous (iv) administration of gadolinium, avoiding some gadolinium-associated risks, such as allergic reactions, renal complications, and gadolinium retention in the brain) [7]. A recent French study investigated the burden of monitoring instruments on 916 IBD patients by administering a visual analog scale questionnaire. IUS resulted in the most tolerated technique in both CD and UC patients, whereas endoscopy demonstrated the lowest acceptability [10]. Factors affecting exam acceptability were bowel cleansing, the fear of complications and abdominal discomfort when performing colonoscopy and venipuncture, and the need for polyethylene glycol for bowel distension when performing MRE [10]. Another study by sMiles et al. showed that patients were more willing to undergo repeated IUS than MRE (99% vs. 91% *p* = 0.012) [11]. Notably, in terms of diagnostic accuracy, both MRE and IUS have a high sensitivity in recognizing the presence of small bowel lesions, although MRE is significantly more sensitive in determining disease extent (80% vs. 70% *p* = 0.027) [12]. IUS accuracy depends on the disease location, with the greatest sensitivity achieved in detecting ileal or left colon lesions (92% and 87%, respectively) and the lowest in the rectum (14%) [13]. Furthermore, a review by Calabrese et al. showed that IUS has a sensitivity and specificity of 79.7% and 96.7%, respectively, for the diagnosis of suspected CD and a sensitivity and specificity of 89% and 94.3%, respectively, for CD monitoring [14]. Table 1 provides a summary of IUS sensitivity and specificity in disease and complications diagnosis compared with MRE.

According to such evidence, IUS is gaining growing importance. Thus, in a recent review on the role of cross-sectional imaging as an alternative to endoscopy for monitoring IBD, Alfarone et al. proposed a flowchart that incorporates IUS in multiple steps of IBD management, from diagnosis to close monitoring after treatment beginning to long-term follow-up [15].

Together with the increasing diffusion of IUS in IBD, the knowledge of sonographic signs of active disease has improved. Several ultrasonographic parameters are nowadays recognized as being associated with intestinal inflammation, including bowel wall thickness (BWT), mural and extramural changes, and variation in bowel wall flow (BWF). Figure 2 shows the normal wall structure. BWT increases as a direct expression of wall inflammation, especially in CD, due to the transmural nature of the disease [7]. Furthermore, a distortion of the normal wall architecture may be present in active disease, as may the presence of increased vascular signals and direct signs of neo-angiogenesis related to disease activity [16]. In particular, the latter is measured with Doppler ultrasonography, and the most frequently used score is the Limberg score, which provides a semi-quantitative characterization, categorizing the vascularization of the wall from 0 to 4 [17]. Finally, alterations of the mesenteric adipose tissue can be observed in IBD. Most frequently, a proliferation of mesenteric adipose tissue can be seen around the affected intestinal loops [16]. Among the different IUS parameters used to assess disease activity, BWT proved to be the most reliable. A threshold of BWT ≥ 3 mm has been established in both CD and UC to distinguish between active and quiescent disease [18]. In a recent meta-analysis and systematic review by Sagami et al., this cut-off showed pooled sensitivity and sensitivity in detecting disease activity of 86.4% and 88.3%, respectively [19].

In the next two subsections, we provide details about the role of IUS in predicting treatment response in CD and UC, respectively. Afterwards, we analyze in a specific subsection the usefulness of IUS in a particular field, the pediatric setting.

### 2.1. Role of IUS in CD

Nowadays, IUS is largely used for CD patients. Since the introduction of TH as a new therapeutic target in CD [4], TH has become one of the main research outcomes. IUS has a great accuracy in detecting TH, showing a high agreement with MRE and mucosal healing [20]. A recent systemic review and expert consensus evaluated treatment response based on IUS parameters. In particular, treatment response was defined as a reduction in BWT > 25%, >2.0 mm, or >1.0 mm associated with a reduction in color Doppler signal, while transmural remission was defined as a BWT ≤ 3 mm with a normal color Doppler signal [21]. In an observational longitudinal study including 218 subjects, Castiglione et al. demonstrated that patients who achieved TH after treatment with an anti-Tumor Necrosis Factor alpha (anti-TNFα) drug showed an improvement in steroid-free clinical remission and a lower rate of hospitalization at one year, compared with only mucosal healing (MH) or no healing (*p* < 0.001) [22].

In order to standardize IUS reporting and research outcomes, some IUS scores have recently been proposed to define inflammatory activity in CD. In 2021, Novak et al. developed the International Bowel Ultrasound Segmental Activity Score (IBUS-SAS) to predict overall disease activity [23]. This index includes the main IUS parameters: BWT, color Doppler signals, Bowel Wall stratification (BWS), and inflammatory fat (i-fat). Despite this score being demonstrated to have excellent reliability, it has yet to be extensively validated, and a cutoff for the definition of active disease needs to be clarified. Moreover in 2021, the Simple Ultrasound Activity Score (SUS-CD) for CD was developed and validated in the same study [24]. This score incorporates an IUS assessment of BWT and color Doppler signals. It showed a good correlation with moderate endoscopic disease activity, defined as a Simple Endoscopic Score (SES-CD) [25] ≥ 7 at ileocolonoscopy (Area under the Curve—AUC 0.88). However, more recently, a Portuguese group compared the accuracy of these two scores and contrast-enhanced ultrasound (CEUS) in predicting inflammatory activity in terminal ileus disease. They obtained that, unlike CEUS parameters, particularly peak intensity, SUS-CD and IBUS-SAS could not correlate accurately with endoscopic activity in the terminal ileum in CD (AUC of 0.80, 0.62, and 0.55, respectively) [26]. Furthermore, in a recent study, IBUS-SAS and SUS-CD significantly correlated with the CD activity index (CDAI). Specifically, in this retrospective study, including about one hundred patients, IBUS-SAS demonstrated a stronger correlation with CDAI with respect to SUS-CD (r = 0.666 vs. 0.486) [27]. More recently, in a large, prospective observational study, Allocca et al. performed IUS on a cohort of 225 CD patients to investigate the role of IUS in predicting a 12-month disease course. They found BWT and BWF to be independent predictors for endoscopic activity and used these two parameters to develop an ultrasound score called the bowel US score (BUSS = 0.75 × BWT + 1.65 × BWF, where BWF = 1 if present or BWF = 0 if absent). A baseline BUSS threshold of 3.52 was identified as an outcome predictor: BUSS values higher than this cut-off reflected persisting endoscopic activity and a negative disease course, with the need for treatment escalation or surgery [28]. In a subsequent study from the same group, the BUSS score proved to modify over time according to endoscopic response/remission [29].

Interestingly, in the last few years, researchers’ attention has focused on IUSs ability to determine an early response to treatment. As it is underlined by Nardone et al. in a recent review on the impact of IUS in IBD management, nowadays IUS is no longer confined to the simple diagnosis; however, it is gaining a more complex role in assessing early response to treatment and supporting clinical decisions [20].

Ripollés et al. demonstrated that an ultrasonographic assessment after 12 weeks of antiTNFα therapy is useful to early predict treatment efficacy. An improvement of IUS parameters (notably BWT and color doppler) at 12 weeks correlated with a one-year response. However, the authors defined treatment response as sonographic and clinical improvement, and endoscopy was not considered in their analysis [30]. In addition, De Voogd et al. found that early reduction of BWT (4–8 weeks after drug induction) predicted an endoscopic response with strong performance in a cohort of CD patients receiving anti-TNFα. Moreover, they confirmed the accuracy of IUS in assessing endoscopic response, identifying 3.2 mm as the best BWT cut-off for detecting endoscopic remission [31]. In a substudy from the Study of Treat to Target Versus Routine Care Maintenance Strategies in CD Patients Treated with Ustekinumab (STURDUST), treatment response and TH were assessed by IUS in a cohort of 77 patients with CD treated with ustekinumab [32]. The IUS parameters considered in the analysis were BWT, BWF, wall stratification, and inflammatory fat. TH was defined as the normalization of all the aforementioned parameters, while the IUS response corresponded to a decrease in BWT of <25%. After 48 weeks of treatment, 24.1% of patients achieved TH. However, IUS response was observed as early as week 4, and there was a good association between IUS response and one-year endoscopic response [32]. Therefore, from a precision medicine perspective, IUS could be a reliable tool to detect early response or failure to ustekinumab, helping to promptly optimize treatment strategies if needed. Furthermore, the recent development of IUS-based ancillary techniques may expand IUS diagnostic and prognostic power. Indeed, the addition of iv contrast agents (CEUS) or oral macrogol solutions (Small Intestine Contrast Ultrasonography, SICUS) increases the ability of standard ultrasonography (B-mode and Doppler parameters) in detecting disease activity, extent, and complications and monitoring treatment response. The most common ultrasound contrast agent used in Europe is SonoVue^®^ [33], which consists of gaseous microbubbles that, once injected, reach the intestinal wall within seconds. It provides an objective and quantitative image of bowel intramural microcirculation and, thus, of disease activity [8]. Dynamic CEUS (D-CEUS) allows one to measure enhancement changes over time and quantify enhancement parameters through specific software analysis programs. The ingestion of oral contrast medium, distending the intestinal lumen, permits a more accurate characterization of the bowel wall and lumen. Therefore, it confers an overall superior sensitivity in detecting small bowel CD lesions and strictures [34].

However, the need for contrast media and specific training still restricts the widespread use of D-CEUS and SICUS. Consistently, data on their predictive role is limited; however, it appears promising.

Some recent studies have investigated the role of CEUS in predicting the response to biologic treatment in CD, and the results agree in underscoring how different CEUS parameters are associated with long-term outcomes [35,36]. In a previous study, our group demonstrated that CEUS may be a predictor of endoscopic and clinical response. We evaluated a cohort of 54 CD patients receiving anti-TNFα and performed D-CEUS at the time of therapy induction and at multiple following time points, aiming to analyze the correlation between D-CEUS and long-term (12 months) endoscopic and clinical response [37]. After two weeks of treatment, the decrease of four D-CEUS parameters (peak intensity—PI; AUC; slope coefficient of wash in—Pw; mean transit time—MTT) was significantly higher in patients who achieved endoscopic response compared with non-responders. In addition, patients who experienced relapse within six months showed a significantly lower reduction in PI and Pw delta, a significant reduction in time to peak (TP) after 12 weeks of treatment, and a significantly lower decrease in AUC at 6 and 12 weeks. Similarly, Quaia et al. investigated the role of CEUS in the early prediction of the long-term response to anti-TNFα therapy. They enrolled 115 consecutive patients who underwent CEUS before starting and after six weeks of treatment. The authors found that percentage changes of the peak enhancement, AUC, AUC during wash-in, and AUC during wash-out were predictors of long-term therapeutic outcomes (*p* < 0.05).

The predictive role of SICUS has been examined in an observational study by Zorzi et al., who enrolled 80 CD patients treated with anti-TNFα drugs who underwent SICUS before and 18 months after treatment induction. Patients were classified as complete responders, partial responders, and non-responders based on SICUS findings. The need for corticosteroids, hospitalization, and surgery one year after the second SICUS was significantly lower in responders compared with partial or non-responders, showing how SICUS can predict long-term outcomes [38].

### 2.2. Role of IUS in UC

Compared with CD, the role of IUS in UC is still less defined. It must be remembered, in fact, that IUS in UC is encumbered by some limitations, notably the typical mucosal rather than transmural involvement, which makes it more difficult to detect bowel lesions with ultrasonographic techniques. In addition, IUS is less sensitive than endoscopy at visualizing rectal activity; therefore, distal disease may be underdiagnosed.

A few studies have focused on the utility of IUS in measuring disease activity and predicting treatment response in UC, and they considered two main parameters: Colonic Wall Thickness (CWT) and Colonic Wall Flow (CWF) [39,40,41,42,43]. The accuracy of IUS in assessing UC disease activity has recently been demonstrated. In 2018, Allocca et al. proposed new non-invasive ultrasound criteria, called Humanitas Ultrasound Criteria (HUC) or Milan Ultrasound Criteria (MUC), capable of identifying active disease (generally defined as a Mayo Endoscopic sub-score ≥ 2) [44]. They prospectively enrolled 53 patients who underwent both an IUS and a colonoscopy. Several IUS parameters were compared with endoscopic activity. On multivariate analysis, only CWT and CWF resulted in independent predictors for endoscopic activity. Therefore, they were used to build up a non-invasive ultrasonography-based score (MUC = 1.4 × CWT in mm + 2 × CWF) to assess and measure disease activity. Notably, a score ≥ 6.3 predicted endoscopic activity with a specificity of 100%. Later, in an external validation cohort, the MUC criteria maintained their good performance, showing an AUC of 0.902, in total agreement with the derivation study [45].

However, data about IUS features predictive of therapeutic response or non-response is still scarce. In 2009, Parente et al. evaluated treatment response in 83 UC patients requiring high-dose steroid therapy. They found that patients with severe ultrasonographic scores at three months from enrollment had a high risk of severe endoscopic activity at 15 months [40]. Later, in the TRUST&UC (TRansabdominal Ultrasonography of the Bowel in Subjects with IBD to monitor disease activity) study, Maaser et al. demonstrated that, after 12 weeks of standard of care therapy (i.e., aminosalicylates, corticosteroids, conventional immunomodulators, and/or biologics as per the physician’s decision), normalization in CWT was highly correlated to clinical response. More specifically, among patients with normalized CWT, 90.5% showed symptomatic response compared to 9.5% of those without symptomatic response (*p* < 0.001) [41]. Moreover, Smith et al. reported that IUS could predict outcomes in patients admitted for UC recurrence. Specifically, CWT, measured within 24 h of admission, was lower in patients who responded to steroids than in those who required rescue therapy (4.6 mm vs. 6.2 mm, *p* = 0.009) [43].

Concerning contrast agents, SICUS is scarcely useful in UC, and, currently, poor data about the use of CEUS are available. Girlich et al. evaluated the role of CEUS in a small cohort of UC patients and compared the results with histological inflammatory activity. Specifically, they considered two CEUS parameters: the percent peak value (Peak), which describes the maximum signal intensity reached during bolus transit, and the TP, which represents the elapsed time from the arrival of the contrast medium to its maximum intensity. They found that increased histological inflammatory activity was related to a decrease in the TP (s)/Peak (%) ratio [46]. Moreover,, in a prospective study, Socaciu et al. examined the ability of CEUS to assess disease activity and treatment response in 65 UC patients in comparison with endoscopy. They first performed CEUS before and three months after treatment initiation and found that AUC was an independent predictor of disease activity both at baseline and during follow-up, with a good correlation with each class of endoscopic activity parameters (*p* < 0.001) [47].

As far as we know, no data about CEUSs role in the early prediction of treatment response in UC is available. More studies are needed to clarify such still-open issues.

Figure 3 provides an example of IUS images showing ultrasonographic parameters of disease activity.

### 2.3. Role of IUS in Pediatric Setting

The IUS role appears crucial, especially in pediatric settings, where repeating an invasive method such as colonoscopy for monitoring disease activity and treatment response is impractical [48]. Endoscopy in children is more uncomfortable because it requires pediatric anesthesiologist assistance and poorly tolerated bowel preparations [48]. On the contrary, the IUS has proven to be the most tolerated monitoring tool, both in the child’s [49] and parent’s perception [50].

As is well known, IUS can accurately detect IBD activity in children [51]. Interestingly, it has recently been proven to predict early response to infliximab induction in pediatric CD patients. In particular, normalization of hyperemia is the most sensitive predictor of treatment response [52]. Moreover, IUS may represent a helpful first-line, noninvasive instrument to evaluate disease activity in children with UC, as it has been demonstrated by Civitelli et al. in their prospective study [53]. They have compared IUS with clinical features in a cohort of 50 pediatric UC patients. They found that the US score showed a strong and significant correlation with the clinical activity of the disease (r = 0.90, *p* < 0.0001) assessed by the Pediatric Ulcerative Colitis Activity Index (PUCAI).

Based on the encouraging data on IUS, the International Bowel Ultrasound Group (IBUS) Pediatric Committee has recently proposed a standardized monitoring algorithm for IBD in pediatric settings by using IUS to help physicians’ decision-making [48] processes More specifically, they suggest performing IUS at baseline (ideally at the time of diagnostic ileocolonoscopy) and whenever a treatment change may be necessary. According to their algorithm, IUS should be repeated after the induction period (at 4–8 weeks) to escalate/optimize therapy if needed. Afterwards, IUS should also be performed together with the ‘treat-to-target’ colonoscopy after the first year and, if the target is achieved, every 3 to 6 months for serial monitoring [48].

## 3. Magnetic Resonance Enterography

MRI, especially MRE, which is performed after administration of oral contrast medium to better visualize intestinal loops, shows some typical characteristics of inflammation in IBD, notably wall thickening, submucosal T2 signal, fat stranding, and ulcerations (which appear as disruption of the inner surface of the bowel wall). To guarantee objective and standardized reports of MRE findings, several MRE activity scores have been developed, considering both sequences with and without iv contrast medium. The main elements included are mural thickness, wall oedema, fat stranding/perimural signal, ulcerations, and contrast enhancement-related parameters.

A recent review by Rimola et al. clearly summarizes the advantages and limitations of such scores [54]. Thus, the MaRIA score, the first and best validated one [55] is time-consuming and requires gadolinium; therefore, a simplified version has been proposed, the so-called sMaRIA, independent from iv contrast medium and quicker to calculate [56]. In 2012, Steward et al. introduced and validated a simple qualitative score, currently known as the London score, based on mural thickness and the mural T2 score. The same authors also proposed an extended version of such an index, the “extended” London score, to include perimural signal and contrast enhancement [57]. These scores are easier to use; however, they were derived by comparing MRE with histological examination of small bowel resected specimens and are not applicable to colon. Nancy scores [58] and Clermont scores [59] differ from the previous ones because they consider diffusion-weighted imaging (DWI) sequences. DWI is based on the Brownian motion of water molecules, which can be restricted due to their interaction with surrounding structures, thus reflecting tissue architecture. It is quantitatively expressed by apparent diffusion coefficient (ADC) values. Nancy score, in particular, has been developed to avoid oral bowel preparation and proved to be a reliable tool to assess colonic inflammation, making it a useful tool in UC [60].

Once MRE accuracy in assessing disease activity was recognized, research focused on its therapeutic implications. MRE parameters potentially able to predict response to medical treatment have therefore been investigated. Due to MREs ability to provide an overall evaluation of the entire gastrointestinal tract and to detect proximal small bowel lesions, it has been especially studied in CD. Data about its role in predicting therapeutic response in UC is still lacking. Moreover, because anti-TNFα antibodies were the first biologics approved for IBD, most of the available results consistently concern this class of drugs.

MRE has been shown to reliably predict clinical outcomes in different studies. Xu et al. demonstrated the correlation between MRE and IUS and clinical disease activity. In particular, they used Spearman’s correlation to assess the relationships between MARIA score (for MRE) and IBUS-SAS score (for IUS) and CDAI and found that both MARIAs and IBUS-SAS had a strong correlation with the clinical activity score (r = 0.485 and r = 0.427, respectively) [61]. Globally, inflammatory features on baseline MRE, such as bowel thickening and increased contrast enhancement, have proven to be associated with better success of medical therapies [62] while penetrating complications are predictors of non-response, leading to surgery [62,63]. More specifically, Buisson et al. demonstrated that, if MRE is performed before anti-TNFα antibodies are first administered, MaRIA score and ADC values predict remission at week 12 after treatment begins [64]. In their study, remission was defined based on clinical and serum parameters (Crohn’s disease activity index and C-reactive protein, respectively). Similarly, in a French multicenter study, Messadeg et al. found that a 25% reduction of the Clermont or MaRIA score at 12 weeks from anti-TNFα first administration could predict corticosteroid-free remission after one year of treatment. In the absence of MaRIA or Clermont scores’ decrease, sustained response was also associated with early improvement of at least two parameters among ulcerations: lymph nodes, ADC, or relative contrast enhancement (RCE) [65]. Clermont and MaRIA scores were significantly associated with 12 months of corticosteroid-free remission in a post-hoc analysis of two previous prospective studies conducted by Buisson et al. [66].

More recently, research outcomes have shifted from clinical symptoms to more objective parameters. In fact, in a treat-to-target strategy, MH has replaced clinical remission as a therapeutic goal. Therefore, MRE predictors of mucosal response currently arouse particular interest. In 2014, Sakuraba et al. published a prospective study on a small group of patients with CD treated with infliximab who underwent both colonoscopy and MRE 1 year after induction. Three years MH, assessed by endoscopy as the gold standard, was considered a reliable sign of maintained response to treatment, and the lack of hyperintensity on DWI was found to be a predictor of sustained response [67]. DWI proved to reliably assess MH in a study by Thierry et al., who applied the Nancy score to detect CD disease activity at baseline and to evaluate response to biologic drugs (infliximab, adalimumab, and vedolizumab). At MRE reassessment after treatment initiation, a total Nancy score < 6 and a segmental Nancy score < 2 defined MH, which was associated with a lower probability of intestinal resection [68].

Moreover, in the last few years, in addition to MH, the concept of TH has emerged, underlying the crucial role of cross-sectional imaging. The combination of endoscopic MH and radiologic TH assessed by MRE showed a statistically significant association with long-term remission in a Polish cohort of CD patients [69]. Moreover, Rimola et al. conducted a prospective study on 58 CD patients requiring anti-TNFα, with the aim of identifying MRE biomarkers of long-term (46 weeks from treatment beginning) TH. Ileal location and presence of creeping fat on baseline MRE were two independent negative predictors, whereas none of the mural parameters (e.g., bowel wall thickness, mural oedema, ulcerations, contrast enhancement) was associated with therapeutic response. Early improvement of the segmental MaRIA score (i.e., a segmental MaRIA score ≤ 10.6 in week 14) had a positive predictive value [70]. More recently, Zhou et al. explored the role of MRE in assessing and predicting TH in CD patients receiving ustekinumab. They retrospectively analyzed a cohort of 37 patients who had undergone MRE at baseline and after 26 weeks of treatment. TH was defined as having a bowel wall thickness ≤ 3 mm without any signs of inflammation at week 26. The subgroup of patients who reached TH, compared with those without TH, had lower bowel wall thickness, fat stranding, and Clermont and MaRIA scores but higher ADC levels [71].

Although many results have been achieved concerning the predictive role of MRE, further studies are needed to solve some still-open issues. Thus, assessing the nature of bowel strictures and evaluating if they may improve after pharmaceutical treatment is not so straightforward. Data from an observational cohort study (CREOLE) of the GETAID group (Groupe d’Etudes Therapeutiques des Affections Inflammatoires Digestives) identified some MRE predictors of response in CD patients treated with adalimumab for small bowel strictures. They found that, on baseline MRE, a stricture length less than 12 cm, a maximal small bowel diameter proximal to the stricture of 18–29 mm, a marked enhancement on the delayed phase, and the absence of fistulas were associated with therapeutic success [72]. Similarly, Amitai et al. retrospectively analyzed 21 patients treated with anti-TNFα for stricturing CD who had undergone MRE before treatment induction. The aim of the study was to assess if MRE features could distinguish between inflammatory and fibrotic strictures, thus predicting a 12-month response to biologic therapies. Their analysis also included DWI parameters. MaRIA and Clermont scores were calculated; however, they showed no correlation with treatment success, while low levels of ADC in DWI were significantly associated with therapeutic failure [73]. Even if current data do not agree about ADC cut-offs, low values of ADC seem to be associated with a fibrotic pattern and consequently a high surgical risk, while strictures with a higher ADC mean may benefit from conservative therapies [74].

In such a context, as underlined in the recent review by Alfarone et al., new MRE-based techniques may be promising, notably magnetization transfer (MT) and motility MRI (mMRI). The first one is a novel tool that distinguishes between protons in free water and large molecules, thus highlighting specific tissues such as collagen [15]. It appears particularly interesting for stricture characterization due to its potential ability to estimate the amount of fibrosis. mMRI, instead, can provide dynamic images of the bowel, reflecting intestinal function and motility [15]. Both of these tools may have a large impact on disease assessment and monitoring; however, further studies are needed to define their roles.

The main MRE and DWI features reflecting active disease and predicting therapeutic response are shown in Figure 4.

## 4. Computed Tomography Enterography and Radiomics

Similarly to MRE, TCE has a high accuracy in IBD diagnosis, being able to early detect mural inflammation, strictures, fistulas, and extra-enteric complications [75]. As for CD, it shows a specificity ranging from 90% to 100% for colon involvement and from 64% to 100% for small bowel lesions [76]. Increased BWT, increased wall contrast enhancement, and fat stranding are signs of active disease detectable with CTE as well as MRE, as shown in Figure 5.

However, an extensive use of CTE in clinical practice is restricted by some limitations, notably radiation exposure, which arouses particular concerns, especially in younger patients who need to undergo frequent disease re-assessments. The presence of valid alternatives among imaging techniques induces physicians to prefer IUS or MRE to regularly monitor disease activity. Thus, CTE is mainly used in emergency settings thanks to its broader availability and accuracy in detecting abdominal complications. Consequently, studies on CTEs role in predicting disease activity are lacking.

Interestingly, promising perspectives come from advanced technologies, namely radiomics. Radiomics consists of extracting quantitative data from medical images to use them to create machine-learning algorithms able to predict outcomes of interest [77]. It found its first applications in the field of oncology; however, in recent years, radiomics has assumed a more significant role even in IBD, standing out as a decision-supporting tool in clinical practice. Currently available studies on IBD have focused mainly on the utility of radiomics in detecting intestinal fibrosis [78,79,80].

In a pilot study, our group first demonstrated the role of radiomics in predicting the risk of surgery at ten years in a cohort of CD patients. We extracted 217 radiomic features from the 93 total lesions identified at the CT scan. Our radiomic model was based on two radiomic parts and could predict surgery with an AUC of 0.83 and sensitivity and specificity, respectively, of 0.72 and 0.90 [81].

On the other hand, to the best of our knowledge, only two studies exist on the role of radiomics in predicting treatment response [82,83]. Chen et al. attempted to predict loss of response to infliximab by constructing a radiomics model based on eight features extracted from pathological CTE lesions in 186 patients naive to biological therapy. The prediction model presented significant discrimination (AUC 0.880) [82]. Likewise, in a retrospective study, Zhu et al. built a radiomics nomogram to early predict MH in a cohort of 106 CD patients after 26 weeks of infliximab treatment. The authors combined five radiomics features extracted by CTE images into a radiomics score (Rad-score) and then integrated them with significant clinical features to develop a radiomics nomogram. In the multivariate analysis, only disease duration confirmed its role as an independent predictor of MH (odds ratio 0.969). The AUC of the radiomics nomogram was 0.880, with 0.760 and 0.833 as sensitivity and specificity, respectively, in distinguishing MH from non-MH [83].

Table 2 schematically synthesizes the main cross-sectional predictive markers. Due to the lack of data about CTE, we only included in the table IUS and MRE parameters.

## 5. Conclusions

Nowadays, cross-sectional imaging techniques are largely integrated into the diagnostic and therapeutic processes of IBD patients, complementing endoscopy and laboratory parameters. Their role in assessing and monitoring disease activity is undoubtedly recognized, especially in CD, due to their ability to examine the entire gastro-intestinal tract and detect TH and extramural involvement. In the last few years, researchers have been trying to define their utility to predict treatment response or failure, addressing the goal of precision medicine with the aim of tailoring therapeutic choices to each patient. In such a context, both IUS and MRE appear particularly promising, and different parameters have been identified as potential predictive markers. CTE is actually more useful in emergency settings or for the detection of IBD complications than in routine IBD assessment; therefore, data about its predictive role are scarce. New technologies, such as CEUS and radiomics, may yield great benefits, and interesting results have already begun to emerge from studies on these topics.

However, data on possible predictive markers is recent and needs to be implemented and validated. We are still far from the application of these new findings in routine clinical practice. Further studies are required to make cross-sectional imaging a reliable guide for IBD physicians aiming to identify the best therapeutic algorithm.

## Figures and Tables

**Figure 1 jcm-12-05933-f001:**
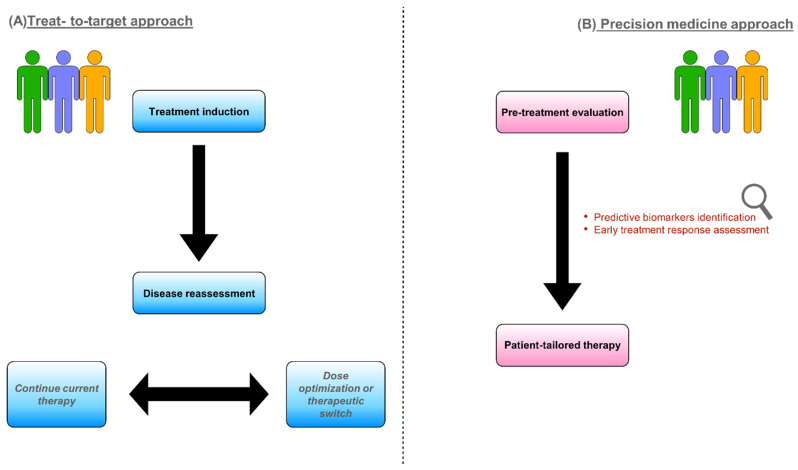
Different models to approach IBD patients requiring biologic treatment: treat-to-target strategy (**A**) and precision medicine (**B**).

**Figure 2 jcm-12-05933-f002:**
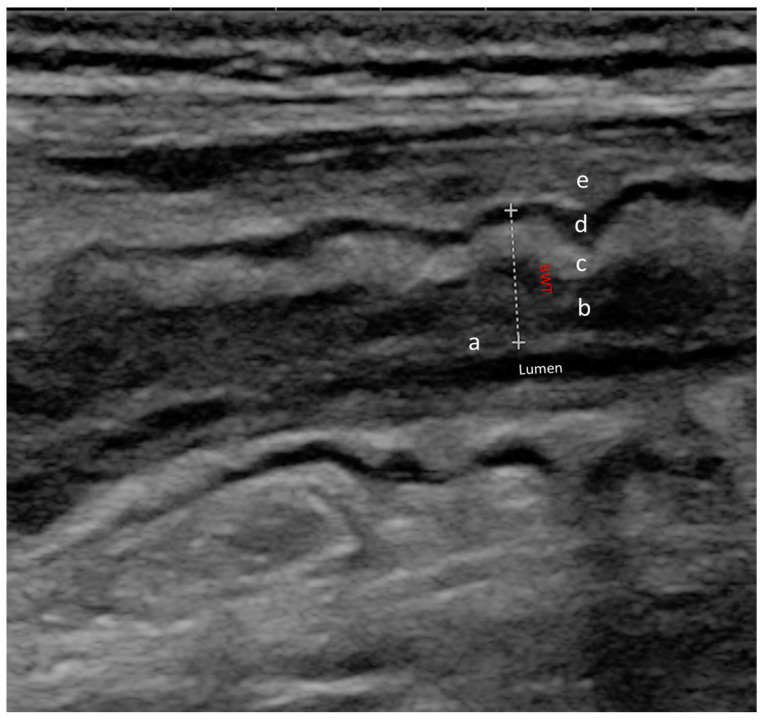
Normal ultrasound bowel wall stratification The bowel wall is composed of five different layers with different echogenic patterns. In order from the innermost to the outermost layer, we can observe: (a) mucosa-lumen interface; (b) muscolaris mucosae; (c) sub-mucosa; (d) muscolaris propria; (e) serosa and extramural structures (peritoneum, fat); BWT (dotted line) is the distance between the lumen-mucosal interface and the serosa.

**Figure 3 jcm-12-05933-f003:**
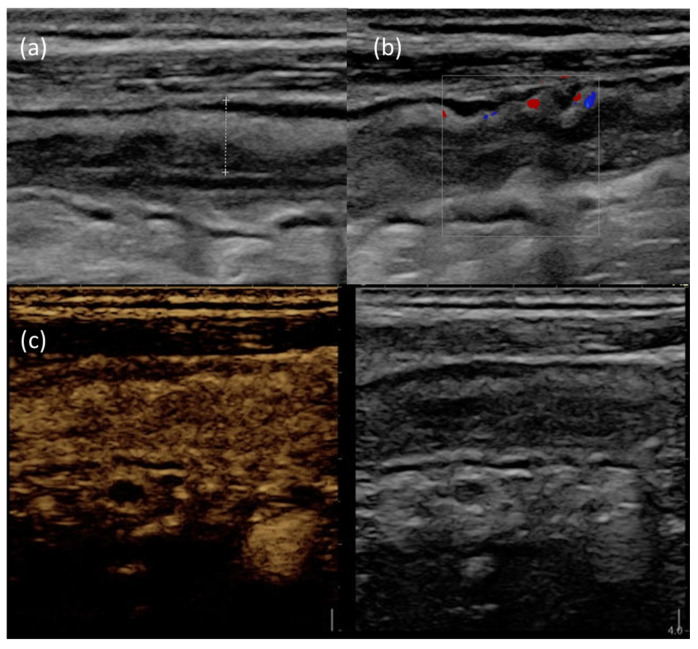
IUS images of the terminal ileum in a patient with active CD. (**a**) B-mode IUS showing increased BWT (dotted line) with preserved wall stratification. (**b**) The use of color doppler highlights increased intramural vascularization. (**c**) After Sonovue^®^ injection, the affected bowel wall shows increased contrast enhancement.

**Figure 4 jcm-12-05933-f004:**
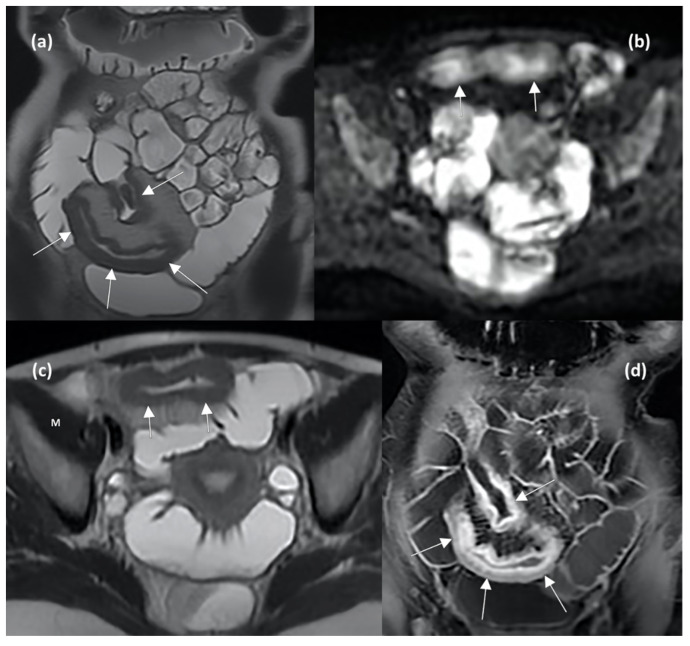
MRE enterography (MRE) images in axial and coronal planes show wall thickening of the distal ileum (arrows in (**a**,**b**)), characterized by hyperintense signal compared to muscle (indicating the presence of oedema, letter M in (**b**)), restricted diffusion in DWI sequence (arrows in (**c**)), and stratified contrast enhancement after gadolinium injection (arrows in (**d**)), indicating active inflammation.

**Figure 5 jcm-12-05933-f005:**
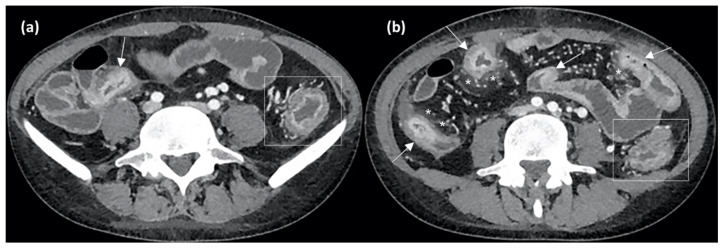
Axial CT enterography (CTE) images after contrast medium injection show wall thickening of the terminal ileum (arrow in (**a**)) and of other small bowel loops (arrows in (**b**)), with stratified contrast enhancement due to hyperdense aspects of the mucosal layer and hypodensity of the submucosal layer, indicating active inflammation. Perienteric fat stranding and local hypervascularization (asterisks in (**b**)) and left colonic wall thickening (rectangles in (**a**,**b**)) are also evident.

**Table 1 jcm-12-05933-t001:** Diagnostic performance of IUS and MRE in IBD. Data are expressed as percentages (95% confidence interval, if available).

	MRE		IUS	
	Sensitivity	Specificity	Sensitivity	Specificity
Disease presence				
Small bowel (regardless of location) [12] Colon [12]	97% (91–99) 47% (31–64)	96% (86–99) 96% (90–98)	92% (84–96) 67% (49–81)	84% (65–94) 96% (90–98)
Disease extent				
Small bowel [12] Colon [12]	80% (72–86) 22% (14–32)	95% (85–98) 93% (87–97)	70% (62–78) 17% (10–27)	81% (64–91) 93% (87–97)
Disease activity				
Small bowel [12] Colon [12]	96% (92–99) 63% (48–76)	83% (68–92) 97% (91–99)	90% (82–95) 66% (51–79)	77% (60–88) 98% (94–99)
Disease complications				
Strictures [7] Fistulas [7] Abdominal [7] abcesses	75–100% 76% 86%	91–96% 96% 93%	80–100% 74% 84%	63–75% 95% 93%

**Table 2 jcm-12-05933-t002:** Cross-sectional imaging predictors of treatment response in IBD.

IUS	MRE
BWT (proposed cut-off <3.2 mm) [20] BWF [21] Wall stratification [21] Inflammatory fat [21] BUSS score [28] D-CEUS parameters: PI, AUC, Pw, MTT [25,26]	Increased mural contrast enhancement [44] Stricture length < 12 cm [46] Small bowel diameter proximal to the stricture is max 29 mm [46] High values of ADC in DWI [47,48] Absence of penetrating complications [46] Scores: MaRIA, Clermont, Nancy [49,50,51,53]

## Data Availability

No new data were created or analyzed in this study. Data sharing is not applicable to this article.

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
