# Peer review of "Predicting Treatment Response in Inflammatory Bowel Diseases: Cross-Sectional Imaging Markers"

_jcm, 2023, doi:10.3390/jcm12185933_

Round 1

Reviewer 1 Report

May be some comment in the pediatric field could be done. 

Correlation among clinical scores ( Simple Colitis Clinical Activity Index (SCCAI), the Mayo Clinic Score (MCS), the Ulcerative Colitis Disease Activity Index (UCDAI), PUCAI for children, CDAI ana PCDAI for children) and and the different imaging test could be find.

Recently, two IUS scores that evaluate inflammatory activity have emerged: the Simple Ultrasound Activity Score for CD (SUS-CD) and the International Bowel Ultrasound Segmental Activity Score (IBUS-SAS). Some comments about these scores are necesary.

Author Response

  1. May be some comment in the pediatric field could be done.

R: following reviewer’s suggestion, we have introduced some comments on the pediatric field, especially concerning the use of IUS, the most diffused and best tolerated diagnostic technique for children. We have introduced a specific subsection to the purpose.         

  1. Correlation among clinical scores (Simple Colitis Clinical Activity Index (SCCAI), the Mayo Clinic Score (MCS), the Ulcerative Colitis Disease Activity Index (UCDAI), PUCAI for children, CDAI and PCDAI for children) and the different imaging test could be found.

R: We thank the reviewer for the suggestion. We have introduced some comments about the correlation of clinical scores and imaging techniques throughout the main text, where appropriate.

  1. Recently, two IUS scores that evaluate inflammatory activity have emerged: the Simple Ultrasound Activity Score for CD (SUS-CD) and the International Bowel Ultrasound Segmental Activity Score (IBUS-SAS). Some comments about these scores are necessary.

R: Thanks to the reviewer for the suggestion. We have included extended comments about SUS-CD and IBUS-SAS scores in the subsection “role of IUS in CD”.

Reviewer 2 Report

In the study by Mignini et al. entitled “Predicting Treatment Response in Inflammatory Bowel Diseases: Cross-Sectional Imaging Markers”, the authors aimed to summarize data about imaging factors that may early predict disease behavior during biological treatment, potentially helping to define more precise and patient-tailored strategies. I have some major comments:

INTRODUCTION
•    The introduction shifts quite abruptly from the discussion of treatment challenges and the role of infliximab to the concept of precision medicine. Consider incorporating a transitional sentence that bridges these two aspects to enhance the flow. This could involve highlighting how the limitations of current approaches have led to the exploration of more individualized treatments, a notion inherent in the precision medicine concept. Providing a more explicit connection between these concepts helps readers follow the progression of ideas more smoothly and logically. Additionally, offering a concise overview of the main sections that the review will cover after this introduction could help readers anticipate the structure and content of the paper.

MAIN TEXT
•    When comparing IUS to other methods like MRE, the text could benefit from using more explicit comparative language. Instead of phrases like "better tolerated by patients," consider specifying how IUS compares to MRE in terms of patient acceptability and convenience. This level of detail enhances the readers' understanding.
•    You should add more studies relevant to your topic recently published (like https://pubmed.ncbi.nlm.nih.gov/35677820/). Specifically, you should compare your results with Alfarone et al.’ review (https://pubmed.ncbi.nlm.nih.gov/35054047/) and write any points you missed in your study.
•    Could you summarize the diagnostic performance of the different imaging modalities in a table using the most recent references?
•    The paragraphs sometimes shift abruptly between discussing various studies and findings. To improve the flow, insert transitional phrases that connect the ideas logically. This could involve summarizing the main point of the preceding discussion and previewing the upcoming topic to create a smoother transition between sentences.

Minor editing of English language is required.

Author Response

In the study by Mignini et al. entitled “Predicting Treatment Response in Inflammatory Bowel Diseases: Cross-Sectional Imaging Markers”, the authors aimed to summarize data about imaging factors that may early predict disease behavior during biological treatment, potentially helping to define more precise and patient-tailored strategies. I have some major comments:

INTRODUCTION

  • The introduction shifts quite abruptly from the discussion of treatment challenges and the role of infliximab to the concept of precision medicine. Consider incorporating a transitional sentence that bridges these two aspects to enhance the flow. This could involve highlighting how the limitations of current approaches have led to the exploration of more individualized treatments, a notion inherent in the precision medicine concept. Providing a more explicit connection between these concepts helps readers follow the progression of ideas more smoothly and logically. Additionally, offering a concise overview of the main sections that the review will cover after this introduction could help readers anticipate the structure and content of the paper.

R: we have implemented the second paragraph, highlighting the connection between current challenges in IBD therapeutic management and the need to precision medicine. In addition, at the end of the introduction, we have provided a more explicit overview of the main text structure.

MAIN TEXT

  • When comparing IUS to other methods like MRE, the text could benefit from using more explicit comparative language. Instead of phrases like "better tolerated by patients," consider specifying how IUS compares to MRE in terms of patient acceptability and convenience. This level of detail enhances the readers' understanding.

R: as it has been suggested by reviewer, factors affecting IUS and MRE acceptability have been more explicitly specified.

  • You should add more studies relevant to your topic recently published (like https://pubmed.ncbi.nlm.nih.gov/35677820/). Specifically, you should compare your results with Alfarone et al.’ review (https://pubmed.ncbi.nlm.nih.gov/35054047/) and write any points you missed in your study.

R: we thank the reviewer for this suggestion. We have thoroughly analyzed Alfarone et al.’ and Nardone et al.’ reviews to identify any points missing in the first version of our paper, which helped us to improve our manuscript. We have updated text and bibliography adding some recent papers that we have previously omitted. We have especially extended the IUS-CD section, including recent evidence about some ultrasonographic scores such as BUSS, IBUS-SAS and SUS-CD, and the MRE section, adding some information about novel promising MRE-based technologies.

  • Could you summarize the diagnostic performance of the different imaging modalities in a table using the most recent references?

R: we have synthesized in Table 1 the diagnostic performance of the two main techniques analyzed in the text, IUS and MRE.

  • The paragraphs sometimes shift abruptly between discussing various studies and findings. To improve the flow, insert transitional phrases that connect the ideas logically. This could involve summarizing the main point of the preceding discussion and previewing the upcoming topic to create a smoother transition between sentences.

R: according to reviewer’s comment, transitional sentences have been introduced to better connect paragraphs make the text easier to read. In addition, in MRE section some paragraphs have been moved to improve text flow.

Round 2

Reviewer 2 Report

Thank you for your revisions.